# Effect of Chlorides Content on the Structure and Properties of Porous Glass Ceramics Obtained from Siliceous Rock

**DOI:** 10.3390/ma15093268

**Published:** 2022-05-02

**Authors:** Alexander Rodin, Anatoly Ermakov, Irina Erofeeva, Vladimir Erofeev

**Affiliations:** Faculty of Architecture and Construction Engineering, National Research Mordovia State University, 430005 Saransk, Russia; anatoly.ermakov97@mail.ru (A.E.); ira.erofeeva.90@mail.ru (I.E.); yerofeevvt@mail.ru (V.E.)

**Keywords:** glass ceramic, foaming, construction material, thermal insulation, siliceous rocks, chlorides, compressive strength, thermal conductivity, thermal analysis

## Abstract

Porous glass-ceramic materials are used in the construction engineering and repair of various objects. The article investigates the method for obtaining porous glass ceramics from siliceous rock with a high calcite content. To obtain samples with an even fine porous structure, a small amount (≤0.386%) of chloride (NaCl, KCl, MgCl_2_·6H_2_O, CaCl_2_) was added to the charge mixture. At the first stage, mechanochemical activation of raw materials was carried out. Siliceous rock, Na_2_CO_3_ and additives (chlorides) were grinded together in a planetary ball mill. The resulting charge was annealed at a temperature of 850 °C. The influence of the type and amount of chloride on the properties of the charge mixture and glass ceramics has been defined by thermal analysis (TA), X-ray diffraction (XRD), scanning electron microscopy (SEM), etc. The chlorides in the charge mixture decreased the calcite’s decarbonization temperature and had an effect on the macro- and microstructure of the material. As a result, samples of glass ceramics with an even finely porous structure in the form of blocks were obtained. The samples consist of quartz, wollastonite, devitrite, anorthoclase and an amorphous phase. On average, 89–90% of the resulting material consists of with small pores. The apparent density of the samples is in the range of 245–267 kg/m^3^. Bending and compressive strength reaches 1.75 MPa and 3.8 MPa, respectively. The minimum thermal conductivity of the modified samples is 0.065 W/(m∙°C). The limiting operating temperature is 860 °C, and the minimum thermal shock resistance is 170 °C. The material has a high chemical stability. They can be used as thermal insulation for some types of industrial and civil facilities.

## 1. Introduction

Porous glass-ceramic materials have many unique properties. They are poor heat conductors, have relatively high strength and chemical stability, are non-inflammable, can be operated at high temperatures, etc. [1,2]. They find a wide utility as thermal insulation in the construction and repair of civil and industrial facilities [3,4].

Raw materials such as glass waste [5,6], slag from metallurgical industries [7], fly ash [1,2,8,9], siliceous rocks [10,11,12], etc., are utilized to obtain porous glass ceramics. The porous structure of the material is obtained in different ways [1,12,13]. One of the most employed is powder foaming. Under this method, a foaming component is added to the charge mixture, which foams the material at a certain temperature. The types of foaming components are diverse: carbonates [1,2,14], carbon mixed with Mn_x_O_y_ [4,15], Na_2_CO_3_ [16], etc.

Porous glass ceramics from siliceous rocks are mainly obtained by the alkaline activation of components [10,11,12]. Diatomite, zeolite-containing tripoli, opoka is mixed with a NaOH aqueous solution of high concentration. The resulting mixture is granulated and annealed. The material comes out in the form of granules. We have obtained porous glass ceramics from siliceous rocks by powder foaming without the use of foaming components. The charge mixture is foamed owing to the zeolite minerals contained in the rock. The method allows to obtain samples in the form of blocks [17]. The main requirement is the joint mechanochemical activation of siliceous (zeolite-containing) rocks and Na_2_CO_3_.

Siliceous rocks have different chemical and mineralogical composition [11,18]. The increased calcite content in the rock makes the macrostructure of glass ceramics uneven. This is due to the superimposition of the effects of the beginning of charge mixture’s melting and decarbonization of calcite. There is little molten in the charge mixture while already a strong gas formation [19]. A partial decrease of the calcite’s decarbonization temperature was achieved by changing the activation mode of the charge mixture: by increasing the activation time, the rotation speed of the mill cups, etc. [18]. We have managed to obtain a fine-pored macrostructure of glass ceramics from siliceous rocks with calcite by introducing amorphous SiO_2_ into the charge mixture as an additive. With the introduction of this additive, the amount of calcite decreases, and the remaining CaCO_3_ decarbonizes at a lower temperature [20]. A similar effect was observed when we had an increase in the amount of Na_2_CO_3_ in the charge mixture [21]. However, following the increase in the amount of amorphous SiO_2_ and Na_2_CO_3_ in the charge mixture composition, the cost of porous glass ceramics increased and some properties of the material deteriorated (the limiting operating temperature fell, chemical resistance decreased, etc.)

The scholarly literature has some evidence on the effect of certain types of additives on the temperature and rate of carbonate minerals’ decomposition during heating. This is known in relation to the decrease in the decomposition temperature of magnesite, dolomite and calcite with different Na salts in the composition [22,23]. Joint annealing of MgCO_3_ and a small amount of LiNO_3_, NaNO_3_, and KNO_3_ sharply accelerated decarbonization [24,25,26]. Joint annealing of brucite, dolomite, serpentite and a small amount of chlorides (NaCl, KCl, etc.), also decreased their decomposition temperature [23,27]. The effect of nitrates, chlorides and some other salts on the decomposition temperature of carbonate minerals is described differently in the literature. Some authors claim that when heated, salts melt, and MgO (CaO) dissolve in the molten and are decarbonized faster [28]. Other researchers claim that molten salts of alkali act primarily as a diffusion medium for CO_2_. The molten prevents the formation of a one-component carbonate impermeable to carbonate ions which facilitates the intensive formation of CO_2_ [29]. So far, we have not been able to find any evidence about the use of chlorides or nitrates as additives that accelerate the decomposition of calcite when obtaining porous glass ceramics from siliceous rocks.

The research objective was to reveal the effect of chlorides (NaCl, KCl, MgCl_2_∙6H_2_O, CaCl_2_) added to mechanochemically activated charge mixture composition (siliceous rock + Na_2_CO_3_) on the structure and properties of porous glass-ceramic materials.

Tasks:To employ thermal analysis (TA) and X-ray diffraction (XRD) to define the effect of additives on phase transformations in the charge mixture during heating and the phase composition of annealed porous glass ceramics;To reveal the effect of additives in the charge mixture composition on the macro- and microstructure of porous glass ceramics;To determine the effect of the content of NaCl, KCl, MgCl_2_·6H_2_O, and CaCl_2_ in the charge mixture composition on the physico-mechanical and thermophysical properties, as well as chemical stability of porous glass ceramics samples.

## 2. Materials and Methods

### 2.1. Materials

Porous glass-ceramic samples were fabricated by mixing siliceous rock, sodium carbonate and additives.

Siliceous rock of the following chemical composition was used: SiO_2_—67.86%; CaO—7.74%; Al_2_O_3_—7.61%; Fe_2_O_3_—1.99%; K_2_O—1.56%; MgO—1.07%; TiO_2_—0.34%; Na_2_O—0.17%; P_2_O_5_—0.15%; SO_3_—0.06%; SrO—0.06%; BaO—0.02%; ZrO_2_—0.01%; V_2_O_5_—0.01%; MnO—0.01%; Cr_2_O_3_—0.01%; LOI—11.32%. The rock’s mineralogical composition: cristobalite—20.5%; heulandite—20.4%; quartz—15.5%; calcite—10.5%; muscovite—13.1%; amorphous phase—20.0%. The siliceous rock was dried to a moisture content of <1%.Sodium carbonate (Na_2_CO_3_) was used to reduce the melting point and foaming of the charge mixture. The purity was ≥99%.To obtain an even macrostructure of pores in glass ceramics, chlorides and their crystallohydrates were used as additives: NaCl, KCl, MgCl_2_·6H_2_O, CaCl_2_. The purity was ≥97%.

### 2.2. Compositions and Fabrication Technology of Samples

Samples of porous glass-ceramic materials were fabricated using the following technology:Mechanochemical activation of raw materials. Siliceous rock, sodium carbonate and additives were being grinded in a planetary ball mill Retsch PM 400 for 35 min (overload inside the crushing cylinder—20G). The required concentration of components was determined by us on the basis of preliminary tests, as well as by the results of previous works [17]. The charge mixture compositions are presented in Table 1.Annealing. The charge mixture obtained after mechanochemical activation was put into metal molds with a size of 120 mm × 120 mm × 260 mm and annealed in a muffle furnace (the molds were previously coated with clay). Annealing schedule: heating to a temperature of 670 °C at a rate of 4.5 °C/min, holding at a temperature of 670 °C for 1 h, heating to a temperature of 850 °C at a rate of 4.5 °C/min, holding at a temperature of 850° C for 30 min; cooling to room temperature inside the furnace.Preparation of samples for testing. The molds with the resulting material were removed from the furnace and dismantled. The resulting porous glass ceramics were sawn into samples of the required sizes and tested.

### 2.3. Analytical Techniques

#### 2.3.1. X-ray Diffraction (XRD)

The X-ray patterns were identified by using an Empyrean PANalytical device (PANalytical, Almelo, The Netherlands) with a PIXcel3D semiconductor detector. The experiment was carried out on grinded glass ceramic samples (fraction < 90 µm). The detector operated in linear scanning mode. Diffraction patterns were identified in CuK_a_ emission in the scan range 2Θ = 4–80°. The step size was 0.0067°/min, the counting time was 150 s. The phase composition of the samples was determined by the Hanawalt method. We used the ICDD PDF-2 database.

#### 2.3.2. Thermal Analysis (TA)

Differential thermal analysis (DTA) and differential thermogravimetry (DTG) of the charge mixture was made using a TGA/DSC1 device (Mettler-Toledo, Greifensee, Switzerland). The charge mixture weighing 20 ± 0.1 mg was put into an alumina crucible with a volume of 150 mcl and compacted. The crucible with the sample was placed in the apparatus and heated from 30 to 850 °C at a rate of 10 °C/min. Images were processed by STARe software (Version SW 10.00, Mettler-Toledo, Greifensee, Switzerland).

#### 2.3.3. Scanning Electron Microscopy (SEM)

SEM micrographs of porous glass-ceramic samples were obtained using the Quanta 200 I 3d apparatus (FEI Company, Hillsboro, OR, USA). Sample scanning mode: pressure—60 Pa, accelerating voltage—30 kV, operating distance—15 mm.

#### 2.3.4. Apparent Density and Porosity

At the first stage, the true density of glass-ceramic materials (*ρ*_0_, g/cm^3^) was found. Then, the cube-shaped samples with 50 ± 5 mm on edge were dried, weighed (*m*_0_, g) and measured (*V*, cm^3^). The samples were put in a cylindrical tank for vacuuming. Water was poured into the tank in such a way that its level in the tank was 20 mm higher than that of the samples. Preliminarily, the water density (*ρ_w_*, g/cm^3^) was determined. Air was pumped out of the tank to a residual pressure equal to 2000 Pa. Testing samples were taken out of the tank after weight stabilization but not less than after 2 h exposure. The water-saturated samples (*m_1_*, g) were weighed indoors. Calculations were done according to the Equations (1)–(4):

Apparent density (*ρ*, g/cm^3^):(1)$$\rho =\frac{m_{0}}{V},$$

Total porosity (*P_t_*, %):(2)$$P_{t}=\frac{\rho_{0}-\rho }{\rho }\cdot 100,$$

Open porosity (*P_o_*, %):(3)$$P_{o}=\frac{m_{1}-m_{0}}{V\cdot \rho_{w}}\cdot 100,$$

Closed porosity (*P_c_*, %):(4)$$P_{c}=P_{t}-P_{o}.$$

The arithmetic mean of the test results of three samples of each composition was taken as the final result.

#### 2.3.5. Bending and Compressive Strength

To define the bending strength, dry samples in the form of rectangular prisms with 120 mm × 30 mm × 30 mm on edges were used. The sample was placed horizontally on two cylindrical supports with a distance of 100 ± 1 mm between them. A cylindrical rod was installed on top along the entire width of the sample at an equal distance from the supports. The diameter of the supports and the rod was 6 ± 0.1 mm. The load on the sample was applied by the rod at a speed of 5 mm/min. The maximum destructive force was taken to be the value at which the sample cracked. The bending strength was calculated according to the standard formula. The arithmetic mean of the test results of three samples of each composition was considered as the final result.

To find out the compressive strength, dry cube-shaped samples of porous glass ceramics with 90 ± 5 mm on edge were tested. The maximum destructive force was identified when the sample was fractured (cracks appeared) or deformed in the surface layers by 10 % of the initial height. The compressive strength of the samples was calculated as the relation of maximum destructive force against the sample’s cross-section area. The arithmetic mean of the test results of five samples of each composition was considered as the final result.

#### 2.3.6. Thermal Conductivity

The thermal conductivity of samples was determined by the probe method using the mobile thermal conductivity meter (MIT–1, LLC “Scientific and production enterprise INTERPRIBOR”, Chelyabinsk, Russian Federation). The testing was done on dry cube-shaped samples with a 90 ± 5 mm on edge. A 6 mm diameter hole with a depth of 50 to 60 mm was drilled in the center of the sample’s side plane. A probe was put into the hole to make readings. The arithmetic mean of the test results of five samples of each composition was taken as the final result.

#### 2.3.7. Thermal Shock Resistance

To identify the values of the thermal shock resistance, cube-shaped samples 50 ± 5 mm on edge were used. The samples were kept in a thermostat at a temperature (*T_T_*) 110 °C ≥ 2 h and then quickly (<10 s) immersed in a water tank. The water temperature (*T_w_*) was 20 ± 2 °C. There, they were kept for 65 ± 5 s. The appearance of cracks on the surface of samples was monitored. The experiment was repeated raising the thermostat temperature by 10 °C until cracks appeared on all samples. The thermal shock resistance of each sample was calculated by the Equation (5):(5)$$\Delta T=T_{T}-T_{w}-10.$$

The arithmetic mean of the test results of four samples of each composition was taken as the final result.

#### 2.3.8. Limiting Operating Temperature

To identify the values of the limiting operating temperature, samples in the form of rectangular prisms 90 mm × 40 mm × 40 mm on edges were used. They were installed vertically in a muffle furnace and tested in the following way: heating to 50 °C less than the set temperature—10 °C/min, heating to the set temperature—2 °C/min, exposure at the set temperature—2 h. During the experiment, the change in the size of the samples was monitored. The limiting operating temperature of the material was determined against the highest test temperature at which the sample sizes changed <1% of the initial values. The experiment was repeated raising the set temperature by 10 °C until the sample sizes changed >1% of the initial values. The arithmetic mean of the test results of three samples of each composition was taken as the final result.

#### 2.3.9. Chemical Stability

To define the values of the chemical resistance of glass-ceramic materials, we tested samples grinded to a fraction of 0.315–0.630 mm. During the experiment, the change in the samples’ weight was controlled after boiling for 3 h in a chemical medium (distilled water, an aqueous solution of 6 N HCl, a mixture of equal volumes of 1 N Na_2_CO_3_ and NaOH solutions). The samples were dried and put into a test jar in an amount of 5 ± 0.0005 g. Then, we dispensed 100 ± 0.5 cm^3^ of reagent into the test jar, connected a backflow condenser to it and boiled the solution. The liquid was drained after boiling, and the sample was washed 5 times with distilled water. The washed sample was drained through a funnel with a paper ash-free filter and placed in a quartz crucible. The crucible with the sample was calcined for 1 h in a muffle furnace at a temperature of 800 ± 10 °C and cooled to 150 °C. The crucible with the sample was cooled to room temperature in a desiccator with CaCl_2_. The samples were weighed, and their weight loss was measured. The arithmetic mean of the test results of two samples of each composition was taken as the final result.

## 3. Results and Discussion

### 3.1. Charge Mixture’s TA

The effect of chlorides on phase transformations in the charge mixture during heating has been established by thermal analysis. The results of differential thermal analysis (DTA) and differential thermogravimetry (DTG) of the charge mixture are presented in Figure 1.

According to Figure 1, the following processes occur in the charge mixture from siliceous rock when heated. The first intensive endothermic effect (Figure 1a) and a significant weight loss of the sample (Figure 1b) were detected at ≈120 °C. According to scholarly literature, sodium hydrosilicates release water in the siliceous charge mixture in this temperature range [30]. When the charge mixture is heated above 300 °C, it triggers the reaction of formation of sodium silicates [11,30]. The reaction is accompanied by an endothermic effect and a significant loss of samples’ weight. The DTG curve of C1 (Figure 1b) shows that the sample is strongly losing its weight until the temperature of ≈550 °C. The next endothermic effect and weight loss of all samples until the temperature reaches ≈650 °C is attributed to the superimposition of the decomposition effects of select minerals of siliceous rock: calcite, muscovite and heulandite. This conclusion is confirmed by the findings in [11,17,30]. Many authors believe that in this temperature range, in the siliceous (zeolite-containing) charge mixture, there is a sealing of surface hydroxyl groups (Si–O–H) in micropores which form water vapor and foam it during the melting of the charge mixture [11,30]. Insignificant endothermic effect and the loss of sample’s weight of C1 with the peak at the temperature ≈680 °C are linked with decarbonization of non-reacting CaCO_3_. The onset of the endothermic melting effect of the charge mixture occurs at ≈680 °C. The temperature at the beginning of the crystallization of the charge mixture sample C1 (the beginning of the exothermic effect) is ≈710 °C. Crystallization is characterized by an intensive exothermic effect with the peak at ≈760 °C.

According to Figure 1, chlorides (NaCl, KCl, MgCl_2_·6H_2_O and CaCl_2_) affect the charge mixture’s phase transformations at >300 °C. By increasing the amount of these chlorides to 0.368% of the charge mixture weight, the intensive loss of the samples’ weight stops at ≈500 °C. This is 50 °C less in comparison with the sample without the additive. With the introduction of chlorides into the charge mixture, the intensity of the endothermic effect and the loss of weight attributable to the decomposition of calcite, muscovite and heulandite decreases. The endothermic effect with a peak at ≈680 °C shifts to the region of lower temperatures (≈665 °C) and is almost imperceptible. According to Figure 1a, the type and amount of chlorides used in the work does not produce an effect on the beginning of charge mixture melting. The increase in the amount of chlorides in the charge mixture to 0.368% shifts the beginning of crystallization by ≈10–20 °C to the region of higher temperatures. The exothermic effects of crystallization are also shifted to the region of higher temperatures and are less intensive compared to samples without additives.

The results of thermal analysis of the charge mixture samples confirmed the positive effect of chlorides on reducing the decomposition temperature of carbonate minerals. Additionally, the effect of chlorides on the crystallization of the charge mixture was revealed.

### 3.2. Grass Ceramics Samples’ XRD

Figure 2 shows the X-ray patterns of samples of annealed glass ceramics. Samples of the control composition (C1) and those modified with chlorides (NaCl, KCl, MgCl_2_·6H_2_O and CaCl_2_) (samples C4, C7, C10 and C13) were tested. For visual clarity, the X-ray patterns are presented in the scan range of 2Θ = 5–45°.

Based on the results of the XRD of the samples (Figure 2), the qualitative phase composition of glass-ceramic materials has been identified. All samples of porous glass ceramics from siliceous rocks consist of a crystalline and amorphous phase. The presence of an amorphous phase in the samples is evidenced on all X-ray patterns by a non-monotonic change in the background (halo) in the scan range from 17 to 37° (2θ). The change of the amorphous halo depending on the type and amount of chlorides in the composition of the material has not been detected. The crystal phase of the control sample (C1) and chloride-modified samples (C4, C7, C10 and C13) consists of quartz [SiO_2_, ICDD: 01-075-8320], wollastonite [CaSiO_3_, ICDD: 01-076-0186], sodium calcium silicate (devitrite) [Na_2_Ca_3_Si_6_O_16_, ICDD: 00-023-0671] and anorthoclase [(Na_0.85_K_0.14_) (AlSi_3_O_8_), ICDD: 01-075-1634].

When modifying the charge mixture with chlorides (NaCl, KCl, MgCl_2_·6H_2_O and CaCl_2_ in an amount up to 0.386% of the total weight), the phase composition of glass ceramics changes slightly. The results of the phase composition of glass-ceramic materials from siliceous rocks are consistent with the scholarly literature findings [17].

### 3.3. Porous Glass Ceramics Macrostructure

To visually confirm the effect of chlorides in the charge mixture composition on the change in the macrostructure of porous glass-ceramic samples, the surface of the latter was scanned. Figure 3 shows scans of the surface of porous glass ceramics samples of the control composition (C1) and samples with NaCl (C2–C4).

According to Figure 3, the sample of the control composition has an uneven macrostructure over the entire surface area. The scan of C1 shows hollows with a diameter of up to 7 mm and channels with a length of ≈10 mm in one cross-sectional plane. Samples of glass-ceramic materials from a charge mixture with NaCl in the amount of 0.092–0.386% have a uniform fine-porous structure (C2–C4). The pore size decreases from ≈ 1 mm to ≈ 0.5 mm against the increase in NaCl from 0.092% to 0.386% in charge mixture composition. The dependence of the macrostructure of porous glass-ceramic samples from the charge mixture with KCl, MgCl_2_·6H_2_O, CaCl_2_ is similar to samples from the charge mixture with NaCl.

### 3.4. SEM Micrographs of Samples

The effect of NaCl in the charge mixture composition on the microstructure of porous glass ceramics samples is shown in Figure 4. The results were obtained by the SEM. A control sample (C1) and samples from a charge mixture with 0.092% and 0.386% of NaCl (C2 and C4) were tested.

According to Figure 4, most of the pores in the control sample (C1, without additives) have the spherical shape. The pores have different diameters (up to 1 mm or more). The surface of the pore walls is not smooth, but rough. The SEM micrograph of C1 shows small pores in the walls connecting large pores with a diameter of up to 0.3 mm. There are many closed and open microscopic pores <50 µm in the pore walls. The SEM micrographs of C2 display pores with a diameter of ≤1 mm. Some pores are connected to each other by wide channels. The pore walls consist of closed micropores. At a higher magnification, microscopic holes with a diameter of ≤20 µm are visible in the pore walls. Sample C2 was obtained from the charge mixture containing NaCl in an amount of 0.092%. The increase in the amount of NaCl in the charge mixture to 0.386% (SEM micrograph C4) causes the appearance of many pores in glass ceramics with a diameter of <0.5 mm. Some pores are linked to each other by holes with a diameter of <0.3 mm. At a higher magnification, microscopic holes are not visible in the pores’ walls. The maximum micropore diameter is less than 20 µm.

Following the completed analysis (Figure 4), it can be stated that small quantities of chlorides in the charge mixture composition produce a significant effect on the microstructure of porous glass ceramics from siliceous rocks. The increase in the amount of NaCl in the charge mixture to 0.386% decreases the pore size and diameter of small pores in the walls connecting large pores. It is known from the scholarly literature that open pores in glass ceramics are a consequence of intensive crystallization of samples [4]. The results of the DTA (Figure 1a) confirm this. The beginning of crystallization in samples with chlorides is shifted to higher temperatures by 10–20 °C, and its intensity is much less.

Additives KCl, MgCl_2_·6H_2_O and CaCl_2_ in the charge mixture produce a similar effect on the microstructure of porous glass ceramics as NaCl does.

### 3.5. Apparent Density and Porosity of Samples

Figure 5 shows dependence graphs of the apparent density and porosity of glass-ceramic samples on the type and amount of additives (NaCl, KCl, MgCl_2_·6H_2_O and CaCl_2_) in the charge mixture composition.

The effect of chlorides in the charge mixture composition on the apparent density of porous glass-ceramic samples is as follows (Figure 5). The apparent density increased from ≈220 kg/m^3^ (C1) to 245–250 kg/m^3^ when NaCl, KCl, MgCl_2_·6H_2_O and CaCl_2_ were introduced into the charge mixture in the amount of 0.092% (C2, C5, C8 and C11). The increase of all these chlorides to 0.386% in the charge mixture composition increased the apparent density of the samples almost linearly to 257–267 kg/m^3^ (depending on the type of additive). A similar influence of all chlorides used in the work on the apparent density of glass ceramics has been found.

According to the findings (Figure 5b), the total porosity of glass-ceramic samples decreases slightly against the increase of chlorides content in the charge mixture composition. In control samples, this indicator was 91.3%. The samples from the charge mixture containing 0.386% CaCl_2_ have the lowest value of total porosity (≈89%). Regardless of the type of additive, 66–69% of the volume of all samples is occupied by open pores. The effect of all chlorides used in the work on the open porosity of glass ceramics is similar. A small amount of additive in the charge mixture (0.092%) decreased the number of open pores in the samples. According to SEM findings (Figure 4), chlorides in small amounts in the charge mixture composition caused the decrease in the diameter of small pores in the walls connecting large pores in the material. When the amount of additives was increased to 0.386% (of the charge mixture’s weight), the open porosity of the samples increased but did not exceed the value of the control composition (Figure 5). The effect may be attributed to the appearance of a system of channels linking adjacent pores inside the sample. The SEM micrograph of C4 (Figure 4) displays that some pores are linked to each other with holes up to 0.3 mm in diameter.

### 3.6. Strength of Samples

The influence of NaCl, KCl, MgCl_2_·6H_2_O and CaCl_2_ admixed to charge mixture on the strength values of porous glass ceramics samples from siliceous rocks is shown in Figure 6.

Following the completed studies (Figure 6a), the bending strength of porous glass-ceramic samples is linearly related to their apparent density. When we increase the content of NaCl, KCl, MgCl_2_ 6H_2_O and CaCl_2_ in the charge mixture composition to 0.386%, the apparent density of the samples increased and, as a consequence, their bending strength did as well. This indicator also depends on the type of additive. For example, samples (C2) from a charge mixture containing 0.092% of NaCl have an average bending strength of 1.2 MPa. In samples (C11) from the charge with the same amount of CaCl_2_, this indicator is 25% higher (1.5 MPa). The apparent densities of samples C2 and C11 are almost equal (≈245 kg/m^3^). Increasing the amount of NaCl and CaCl_2_ in the charge mixture to 0.386% leads to the increase in bending strength of porous glass ceramic samples (C4 and C13) to 1.35 and 1.75 MPa, respectively. C13 samples are almost 30% stronger than C4 samples with a slight difference in apparent density. The bending strength of samples from the charge mixture containing equal amount of chloride increased with respect to the type of additive in the following sequence NaCl→KCl→MgCl_2_·6H_2_O→CaCl_2_. The effect can be attributed to the change in the phase composition of the samples. According to the X-ray patterns (Figure 2), the sample of glass ceramics from the NaCl charge mixture has the smallest amount of wollastonite, and from the CaCl_2_ charge mixture, the largest. The bending strength of glass-ceramic materials containing different amounts of wollastonite was investigated in [31]. We also have to draw attention to the different patterns of the crystallization process in samples from charge mixture containing different chlorides (Figure 1). In samples from the charge mixture with NaCl, crystallization takes place intensively and at a lower temperature. Crystallization in samples with CaCl_2_ is less intensive with a maximum at a higher temperature.

According to the research findings (Figure 6b), the compressive strength of porous glass-ceramic samples from the charge mixture with chlorides also increases linearly against the increase in their apparent density. The highest value belongs to samples C10 and C13 (≈3.8 MPa). The average density of the samples is in the range of 256–267 kg/m^3^. The samples of the control composition (C1) have the lowest compressive strength ≈2.2 MPa and apparent density ≈220 kg/m^3^. Following the findings for the bending strength, the same relationship between the increase in compressive strength of samples from the charge mixture with an equal amount of chloride and the type of additive (NaCl→KCl→MgCl_2_·6H_2_O→CaCl_2_) remains. The introduction of chloride additives into the siliceous charge mixture with sodium carbonate allowed to obtain porous glass-ceramic materials which having equal apparent density, surpass foam glass and glass ceramics from industrial waste in strength values [1,2,10,11,12].

### 3.7. Thermal Conductivity of Samples

The results of thermal conductivity of porous glass-ceramic samples are shown in Figure 7. The results are shown in relation to the apparent density of the samples.

Based on the results of the completed analysis (Figure 7), a linear dependence of the samples’ thermal conductivity on their apparent density has been revealed. The dependence is valid for glass ceramic samples with an apparent density from 210 to 270 kg/m^3^. It can be written down in the following Equation (6):(6)$$\lambda =17.1\cdot 10^{-5}\cdot \rho +0.023,$$
where: λ—thermal conductivity (W/(m∙°C)),

ρ—apparent density of a dry material (kg/m^3^).

Correlation coefficient (R^2^) ≈ 0.92.

According to experimental findings (Figure 7), the control samples (C1) have the lowest thermal conductivity equal to an average of 0.06 W/(m∙°C). The apparent density of the samples is 220 kg/m^3^. According to Figure 3, the macrostructure of the sample C1 is uneven. The lowest thermal conductivity of porous glass ceramics with a uniform fine-porous structure is in samples from the charge mixture containing 0.092% NaCl, KCl, MgCl_2_·6H_2_O and CaCl_2_. It is equal on average to 0.065–0.066 W/(m∙°C) with an apparent sample density from 245 to 249 kg/m^3^. Samples from the charge mixture with the maximum amount of chlorides (0.386%) have the highest value of thermal conductivity which is ≈0.068 W/(m∙°C). The apparent density of these samples is from 262 to 267 kg/m^3^. The results obtained correlate with the findings of previous studies [3].

### 3.8. Thermal Shock Resistance

In order to expand the scope of application of the developed materials, their thermal shock resistance has been defined. Porous glass ceramics with high values of this parameter can be used as thermal insulation of industrial equipment [3,32]. The results of experimental testing are presented in Figure 8.

After analyzing the values of Figure 8, it was found that the thermal shock resistance of porous glass-ceramic samples decreased against the increase in the amount of chlorides in the charge mixture (NaCl, KCl, MgCl_2_∙6H_2_O and CaCl_2_) to 0.184% or more. The lowest values of thermal shock resistance on average equal to 170 °C in samples C3, C4, C6, C7, C10, C12 and C13. The highest thermal shock resistance of the control samples is ≈190 °C. It was not possible to find the influence of the phase composition of the material on the samples’ thermal shock resistance. According to Figure 2, the effect of chlorides in the charge mixture on the phase composition of glass ceramics is insignificant. The decrease in the samples’ thermal shock resistance as response to the increase in the amount of chloride in the charge mixture can probably be attributed to the ordering of the structure of the material. According to Figure 3 and Figure 4, sample C1 consists of large macropores up to 7 mm in diameter and small pores in the walls connecting large pores up to 0.3 mm. Increasing the content of additive in samples decreases the pore diameter (<1 mm), and the holes are practically not visible on SEM micrographs. Probably, the initial temperature retained longer inside the material with a uniform fine-pored structure. A large temperature variation was formed between the surface of the sample and its middle part. As a result, the thermal shock resistance of the samples fell. The findings are consistent with the results of other researchers. The relationship between thermal shock resistance and porosity of glass ceramics is described in detail in the scholarly literature [4].

The obtained values of thermal shock resistance of porous glass-ceramic materials are almost identical to those obtained from industrial waste [32]. The developed materials can be recommended as thermal insulation for some types of industrial equipment [3,32].

### 3.9. Limiting Operating Temperature

One of the main criteria for the use of porous material in the thermal insulation of industrial equipment is the limiting temperature of its operation. The effect of the type and amount of additive (NaCl, KCl, MgCl_2_·6H_2_O and CaCl_2_) in the charge mixture composition on the limiting operating temperature of glass-ceramic samples from siliceous rocks is shown in Figure 9. The parameter value was determined by measuring the size of samples after holding them for 2 h at a given temperature.

Based on the completed experimental studies, it was found that the addition of NaCl, KCl, MgCl_2_·6H_2_O and CaCl_2_ to the charge mixture in the amount of up to 0.386% (of the charge mixture weight) had almost no effect on the limiting operating temperature of the samples (Figure 9). The developed porous glass-ceramic materials can be operated at temperatures under 860 °C. The residual sizes of the samples after holding them for 2 h at this temperature were more than 99% of their initial values.

As mentioned above, chlorides were introduced into the siliceous charge mixture with a high calcite content to obtain a uniform fine-pored structure of glass ceramics. A similar effect can be achieved by increasing the amount of amorphous SiO_2_ (for example, diatomite) and Na_2_CO_3_ [21]. However, the limiting operating temperature of such glass-ceramic materials rarely exceeds 800 °C [20]. Moreover, in terms of the limiting operating temperature, porous glass-ceramic materials from siliceous rocks (modified with chlorides) are significantly superior to foam glass, as well as porous glass-ceramics from siliceous rocks obtained by alkaline charge mixture activation [10,11,12,19,30]. The developed materials can be used as thermal insulation for some types of industrial equipment.

### 3.10. Chemical Stability

The effect of the type and amount of chlorides in the charge mixture composition on the chemical stability of porous glass-ceramic samples is shown in Table 2. The criterion for estimating this parameter was the weight loss of grinded samples (fraction 0.315–0.63 mm) after boiling them for 3 h in various chemical media.

According to the findings (Table 2), the developed glass-ceramic materials from siliceous rocks have high chemical stability in water. This indicator decreases slightly in samples from the charge mixture with chloride. After boiling for 3 h, the samples from the charge mixture with the maximum amount of additive (NaCl, KCl, MgCl_2_·6H_2_O and CaCl_2_) 0.386% lost about 1% in weight on average. The result obtained makes it possible to recommend the developed materials for use in wet conditions.

The effect of chlorides on the chemical stability of glass ceramic samples in an aqueous solution of HCl (6 N) was found. The stability increased slightly or was almost equal to the value of the control composition when we increased the chlorides content in the charge mixture to 0.092% and 0.184% (Table 2). After the increase in the amount of the additive to 0.386%, the chemical stability of the samples decreased. Weight loss of samples containing NaCl (C4) and KCl (C7) in the charge mixture was >5%. It is known from the scholarly literature that different minerals have different resistance to the action of acids [33]. According to Figure 2, glass ceramics samples C4 and C7 contain the largest amount of devitrite. This probably led to the decrease in the chemical stability of the samples in an aqueous solution of HCl (6N).

The stability of glass ceramics from siliceous rocks to the alkaline solutions exposure (Na_2_CO_3_(1 N) + NaOH(1 N)) decreased slightly due to the increase of NaCl, KCl, MgCl_2_·6H_2_O and CaCl_2_ (Table 2) in the charge mixture composition. With the maximum additive content in the charge mixture (0.386%), the weight loss of glass ceramic samples (C4, C7, C10 and C13) after boiling in an alkaline solution for 3 h increased by 1–1.5% on average. The effect may also be attributed to a slight change in the phase composition of glass ceramics caused by the type and amount of additive in the charge mixture (Figure 2).

Following the results of the completed studies, high chemical stability of glass-ceramic samples from siliceous rocks has been revealed. The values obtained are higher or equal to the values of analogues [34,35]. Even if we consider a slight decrease in the chemical stability of select samples from the charge mixture with chlorides. The developed materials can be used as thermal insulation for some types of pipelines, industrial installations, etc.

## 4. Conclusions

Porous glass-ceramic materials were obtained from siliceous rock with a high calcite content. Mechanochemical activation of the rock, sodium carbonate and an additive was carried out in a planetary ball mill. Chlorides (NaCl, KCl, MgCl_2_·6H_2_O and CaCl_2_) were used as additives to obtain glass-ceramic samples with an even fine-pored structure. After activation, the charge mixture was annealed at a temperature of 850 °C. The influence of the type and quantity of additives on the properties of the charge mixture and glass ceramics has been identified by thermal (TA) and X-ray diffraction (XRD) analysis, scanning electron microscopy (SEM), etc.

Main conclusions:A small amount of chloride produces a significant effect on the phase transformations in the charge mixture obtained from siliceous rock. The increase in the amount of additive in the charge mixture to 0.368% has accelerated the formation of sodium silicates and decreased the calcite’s decarbonization temperature. The peak of the exothermic effect shifted to higher temperatures by ≤40 °C.Samples of glass ceramics with a uniform fine-porous structure from siliceous rock with a calcite content (10.5%) have been obtained by modifying the charge mixture with chlorides (NaCl, KCl, MgCl_2_·6H_2_O and CaCl_2_) in the amount of 0.096 to 0.368%. The pore diameter in the material decreased to ≈0.5 mm after the increase in the amount of additive. Glass ceramic samples consist of quartz, wollastonite, devitrite, anorthoclase and amorphous phase. The additives used (chlorides) have a negligible effect on the phase composition of the samples.The developed porous glass ceramic has an apparent density of 245–267 kg/m^3^; bending and compressive strength up to 1.75 MPa and 3.8 MPa, respectively; thermal conductivity 0.065–0.068 W/(m∙°C); thermal shock resistance 170–180 °C; limiting operating temperature up to 860 °C; and high chemical stability.Comparing some indicators, the obtained materials are superior to foam glass and other analogues. They can be used as thermal insulation for some types of industrial and civil facilities.

## Figures and Tables

**Figure 1 materials-15-03268-f001:**
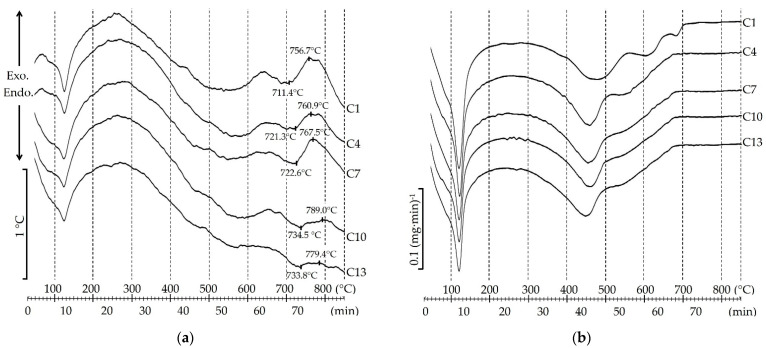
DTA (**a**) and DTG (**b**) curves of charge mixture samples.

**Figure 2 materials-15-03268-f002:**
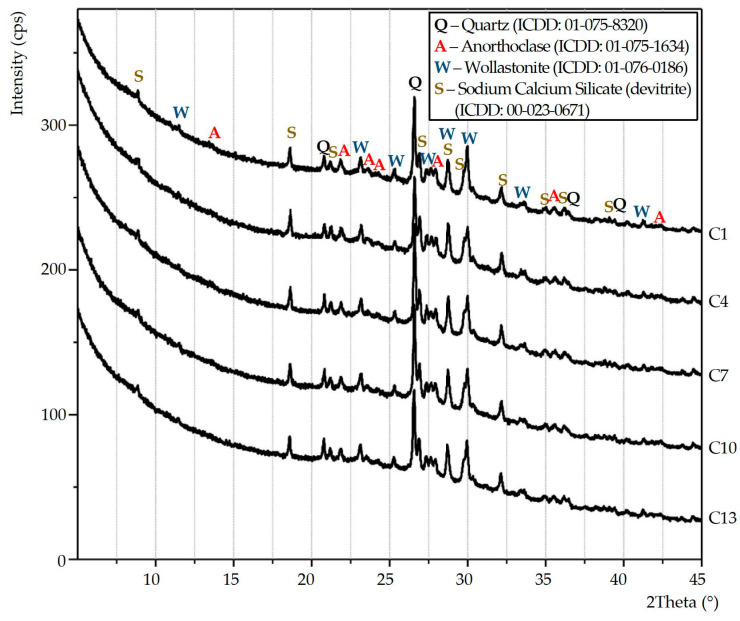
X-ray patterns of glass ceramics samples.

**Figure 3 materials-15-03268-f003:**
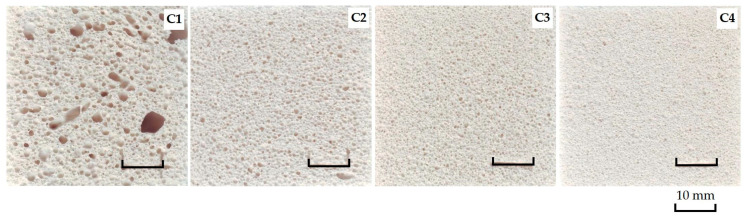
Scan of the surface of porous glass-ceramic samples.

**Figure 4 materials-15-03268-f004:**
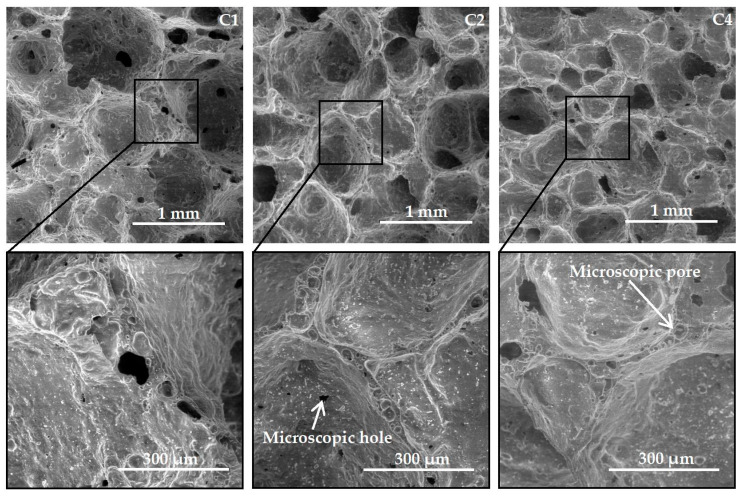
SEM micrographs of samples.

**Figure 5 materials-15-03268-f005:**
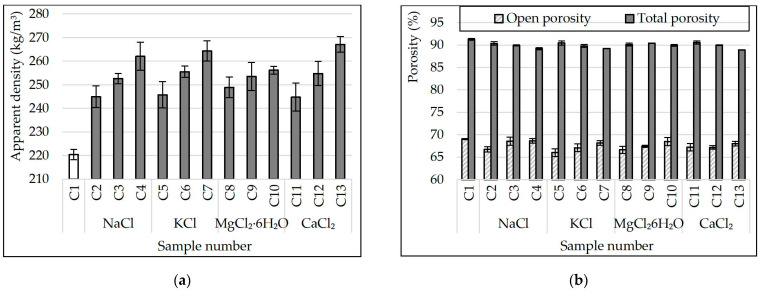
Apparent density (**a**) and porosity (**b**) of samples.

**Figure 6 materials-15-03268-f006:**
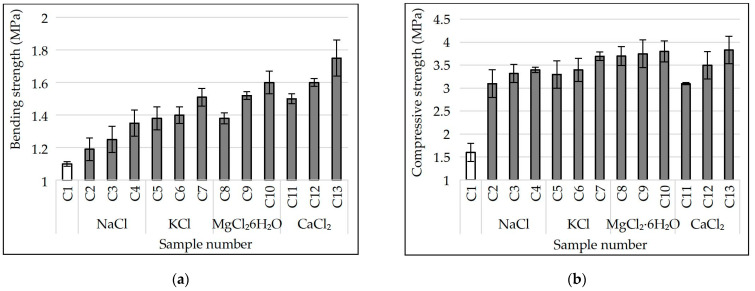
Bending (**a**) and compressive (**b**) strength of samples.

**Figure 7 materials-15-03268-f007:**
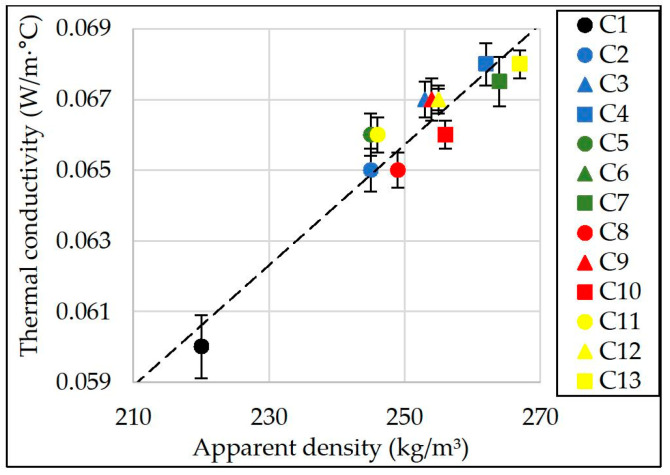
Thermal conductivity of samples.

**Figure 8 materials-15-03268-f008:**
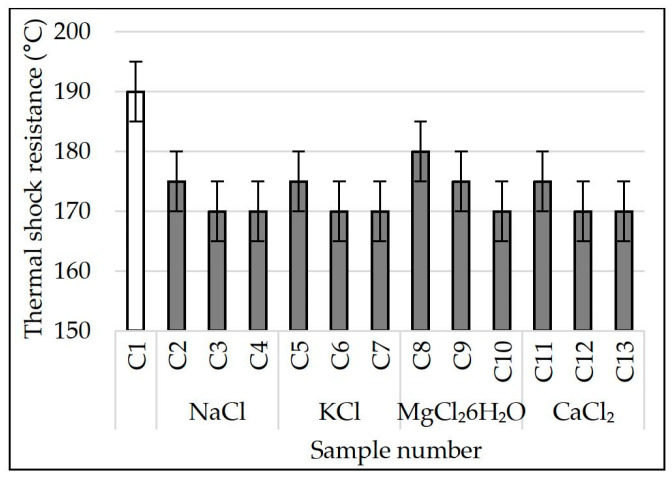
Thermal shock resistance of samples. White square – without chlorides; Gray squares – with chlorides.

**Figure 9 materials-15-03268-f009:**
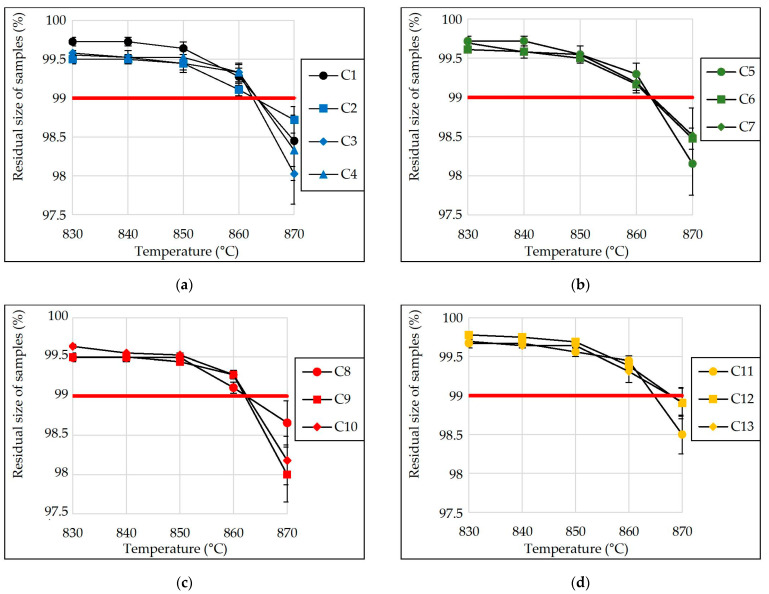
Residual size of samples after exposure to the set temperature for 2 h (samples from charge mixture containing: (**a**) no additive and with NaCl; (**b**) with KCl; (**c**) with MgCl_2_·6H_2_O; (**d**) with CaCl_2_; C1–C1_3_—numbers of compositions).

**Table 1 materials-15-03268-t001:** Charge mixture’s compositions.

No. of Composition	Charge Mixture’s Compositions, %					
	Siliceous Rock	Na_2_CO_3_	NaCl	KCl	MgCl_2_·6H_2_O	CaCl_2_
C1	81.6	18.4	–	–	–	–
C2		18.308	0.092	–	–	–
C3		18.216	0.184	–	–	–
C4		18.032	0.368	–	–	–
C5		18.308	–	0.092	–	–
C6		18.216	–	0.184	–	–
C7		18.032	–	0.368	–	–
C8		18.308	–	–	0.092	–
C9		18.216	–	–	0.184	–
C10		18.032	–	–	0.368	–
C11		18.308	–	–	–	0.092
C12		18.216	–	–	–	0.184
C13		18.032	–	–	–	0.368

**Table 2 materials-15-03268-t002:** Change in the samples’ weight after boiling in chemical media for 3 h.

Composition No.	Change in the Samples’ Weight after Boiling in Chemical Media for 3 h, % *		
	H_2_O	6 N HCl Solution	1 N Na_2_CO_3_ Solution + 1 N NaOH Solution (1:1)
C1	0.34	4.34	7.34
C2	0.65	3.61	7.67
C3	0.76	4.48	7.97
C4	1.05	5.04	8.07
C5	0.89	4.29	8.19
C6	0.98	4.50	8.36
C7	1.21	5.34	8.81
C8	1.10	4.10	7.55
C9	1.07	4.29	8.05
C10	1.05	4.72	8.57
C11	1.22	3.86	7.86
C12	0.46	3.81	8.10
C13	0.91	4.49	8.45

* The differences in the test results of the samples of each composition did not exceed 5% of the average value.

## Data Availability

All data are freely available.

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
