# Peer review of "Effect of Chlorides Content on the Structure and Properties of Porous Glass Ceramics Obtained from Siliceous Rock"

_materials, 2022, doi:10.3390/ma15093268_

Round 1
Reviewer 1 Report
The research proposed a study to give the effect of chlorides content on the structure and properties of porous glass ceramic. However, I think that this paper is not acceptable in its present form. There were some language errors and a lack of scientific explanations in this article. Moreover, the literature survey was slightly deficient. Therefore, I would recommend major revisions to this current version of the article.
1-Introduction: In this section, there are not enough reference citations and sufficient descriptions, which leads me to think that this paper is lacking in the literature review.
2-Language: Some of the language errors in this manuscript should be revised.
3-Redraw the Fig.1 due to the unclear showing of the explanations of the effect of chloride additives on DTA and DTG curves. Besides, the description of Fig.1 should be rewritten. The author should first give the overall graphic description, and then give the impact caused by the addition of chloride.
4-Fig.2: There is no clear difference between different samples. Local magnification may be helpful to the author's conclusion.
5- The discussion in sections 3.2, 3.3, and 3.4 only compares the different amounts of one kind of chloride. I wonder if there is a difference in samples with different chlorides, and which kind of chlorides has a more obvious effect.
Author Response
Cover Letter
Thank you very much for your comments. We have tried to correct all of them.
Our answers:
- Manuscript text changed. Lines 65–80, 607–622.
- Manuscript text changed. Lines 25, 26, 121, 123, 194-196, 202, 203, 208, 209, 214, 215, 450–452, 466, 467, 470, 503, 504, 579–581, 590–591, 594, 595, 597, 599, 602, 606–608, 610, 618, 710, 722, 723, 726, 736, 739.
- Figure 1 modified.
Manuscript text changed. Lines 260–297.
- Manuscript text changed. Lines 390, 391.
- Manuscript text changed. Lines 406-408, 471, 472.

Reviewer 2 Report
The paper presents the influence of various chlorides on the microstructure of foamed glass-ceramics based on siliceous rocks.
The authors are asked to shorten the description of the obtained results (sections 3.1 to 3.10). It is not needed to mention all the numbers in the text, but only the trends and some specific data if they are standing out or are important for some reason.
Specific comments:
- Experimental: “The total mass fraction of the basic material ≥ 99 %.” And “The concentration of the main substance in the additives is ≥ 97 %.” These sentences probably refer to purity, so state “the purity is…”
- “The maximum destructive force was taken to be the value at which the height of the sample decreased by 10 % from the initial value.” Why did you take 10% value? Usually for brittle materials like ceramic/glass foams, the highest compressive force is attained before the material starts to crush which is usually before 10% deformation.
- State exactly (model, brand, producer) which apparatus was used to determine thermal conductivity. For your info, the needle probe method is not the best choice for brittle materials. Do you maybe have a reference material (e.g., commercial foamed glass with known conductivity) to make evaluation of the method? The best methods for assessing thermal conductivity are HotDisk (small samples) and heat-flow meter (large samples, 20x20 cm).
- 1. DTA and DTG: change to TA and TG curves as it is easier to see the mass changes.
- DTA and DTG curves look very similar, maybe decrease the number of curves to only show the highest chloride additions and reference. The explanation of Fig. 1 results should be decreased by at least 66%.
- “chlorides produce a significant effect on charge mixture’s phase transformations” mark these changes in Fig. 1.
- 2 XRD: the change in intensity is very small. The conclusion could be that addition of chlorides doesn’t influence the phase composition much.
- Microstructures: the change in the color is not important, so remove the related text.
- “the shape of a ball” = spherical shape
- “surface of the pore walls is uneven” = is not smooth, but rough. This is related to crystallization.
- “through holes” = small pores in the walls connecting large pores
- “shifted by 10–20 °C to the region of high temperatures” = shifted to higher temperatures (correct in the whole article)
- “According to Figures 3 and 4, a slight increase in the apparent density of the samples caused the appearance of the material with a uniform fine-pored structure.” This is not the real cause. The change is related to the crystallization, surface tension, and similar changes when adding chlorides. The C1 sample with 250 kg/m3 (e.g., prepared at a lower density) would not look like the samples with chlorides.
- “It is impossible to recommend such a material as thermal insulation.” This is a hard statement, few large pores in the material will not make a difference in use. The more uniform chloride samples have better mechanical performance, but for thermal insulation, conductivity is the most important.
- “thermal stability” = thermal shock resistance
- “The parameter value was determined by weighing the size”. The size is measured not weighted.
- 9: you can change y-axis scale to 98‒100% or similar, to show also the datapoints <99%.
- “The developed materials can be used as insulation for pipelines, industrial installations, etc”. “They can be used as thermal insulation for industrial and civil facilities.” In general, open porosity is unwanted in thermal insulation since it can give rise to accumulation of water in the material. To prevent this, insulation materials are made hydrophobic. How do you comment on this? Is the developed material really very suitable?
- In conclusion you state that “The additives used (chlorides) have a negligible effect on the phase composition of the samples.”, while you use discuss in a long text about phase composition. Therefore, the description of the results need to be shortened.
Author Response
Thank you very much for your comments. We have tried to correct all of them.
Our answers:
- Manuscript text changed. Lines 121, 123.
- Manuscript text changed. Lines 194–196. The samples did not crack during testing. They were deformed in the surface layers the same way as in the testing of foam glass samples. The test procedure is taken from DIN EN 13167 and Russian National Standard 33949-2016.
- Manuscript text changed. Lines 202, 203.
- Figure 1 modified.
- Manuscript text changed. Lines 260–297.
- Figure 1 modified.
- Manuscript text changed. Lines 390, 391.
- Manuscript text removed.
- Manuscript text changed. Lines 450.
- Manuscript text changed. Lines 451.
- Manuscript text changed. Lines 451, 452, 466, 467, 503, 504, 602.
- Manuscript text changed. Lines 470, 726.
- Manuscript text removed.
- Manuscript text removed.
- Manuscript text changed. Figure 8 modified. Lines
- Manuscript text changed. Lines 25, 208, 209, 214, 215, 579–581, 590, 591, 594, 595, 597, 599, 606–608, 610, 736.
- Figure 9 modified.
- Manuscript text changed. Lines 26, 612, 710, 739. The developed materials can be used as thermal insulation for some types of industrial equipment (Thermal insulation of furnaces, etc.).
- Manuscript text changed. Lines 390, 391.

Round 2
Reviewer 1 Report
Accept in present form